# Multiparametric Analysis of Longitudinal Quantitative MRI Data to Identify Distinct Tumor Habitats in Preclinical Models of Breast Cancer

**DOI:** 10.3390/cancers12061682

**Published:** 2020-06-24

**Authors:** Anum K. Syed, Jennifer G. Whisenant, Stephanie L. Barnes, Anna G. Sorace, Thomas E. Yankeelov

**Affiliations:** 1Department of Biomedical Engineering, The University of Texas at Austin, Austin, TX 78712, USA; anum.syed@utexas.edu; 2Department of Medicine, Vanderbilt University Medical Center, Nashville, TN 37232, USA; j.whisenant@vanderbilt.edu; 3Oden Institute for Computational Engineering and Sciences, The University of Texas at Austin, Austin, TX 78712, USA; steph.eldridge82@gmail.com; 4Department of Biomedical Engineering, The University of Alabama at Birmingham, Birmingham, AL 35294, USA; asorace@uabmc.edu; 5Department of Radiology, The University of Alabama at Birmingham, Birmingham, AL 35294, USA; 6O’Neal Comprehensive Cancer Center, The University of Alabama at Birmingham, Birmingham, AL 35294, USA; 7Department of Diagnostic Medicine, The University of Texas at Austin, Austin, TX 78712, USA; 8Department of Oncology, The University of Texas at Austin, Austin, TX 78712, USA; 9Livestrong Cancer Institutes, The University of Texas at Austin, Austin, TX 78712, USA

**Keywords:** diffusion-weighted MRI, dynamic contrast-enhanced MRI, immunohistochemistry, habitat imaging, intratumoral heterogeneity

## Abstract

This study identifies physiological tumor habitats from quantitative magnetic resonance imaging (MRI) data and evaluates their alterations in response to therapy. Two models of breast cancer (BT-474 and MDA-MB-231) were imaged longitudinally with diffusion-weighted MRI and dynamic contrast-enhanced MRI to quantify tumor cellularity and vascularity, respectively, during treatment with trastuzumab or albumin-bound paclitaxel. Tumors were stained for anti-CD31, anti-Ki-67, and H&E. Imaging and histology data were clustered to identify tumor habitats and percent tumor volume (MRI) or area (histology) of each habitat was quantified. Histological habitats were correlated with MRI habitats. Clustering of both the MRI and histology data yielded three clusters: high-vascularity high-cellularity (HV-HC), low-vascularity high-cellularity (LV-HC), and low-vascularity low-cellularity (LV-LC). At day 4, BT-474 tumors treated with trastuzumab showed a decrease in LV-HC (*p* = 0.03) and increase in HV-HC (*p* = 0.03) percent tumor volume compared to control. MDA-MB-231 tumors treated with low-dose albumin-bound paclitaxel showed a longitudinal decrease in LV-HC percent tumor volume at day 3 (*p* = 0.01). Positive correlations were found between histological and imaging-derived habitats: HV-HC (BT-474: *p* = 0.03), LV-HC (MDA-MB-231: *p* = 0.04), LV-LC (BT-474: *p* = 0.04; MDA-MB-231: *p* < 0.01). Physiologically distinct tumor habitats associated with therapeutic response were identified with MRI and histology data in preclinical models of breast cancer.

## 1. Introduction

Intratumoral heterogeneity plays a fundamental role in tumor progression, treatment resistance, and disease recurrence [1]. Tumors contain phenotypically and genetically diverse cancer cell populations of potentially varying proliferation rates and therapeutic sensitivities [1,2]. Additionally, the tumor microenvironment is remodeled as cancer cells recruit nonmalignant stromal cell populations (e.g., immune cells, fibroblasts, and vascular endothelial cells), thereby creating spatial variation within the tumor [3,4]. The resulting spatial distribution of cancer and stromal cells leads to local alterations in architecture of the microenvironment and the development of physiologically distinct subregions [4,5,6]. For example, high cancer cell proliferation and metabolism results in areas of acidic pH, and disparate vessel recruitment leads to regional variation in oxygen concentration across the tumor [4,5]. These tumor subregions can then exert selective pressures onto the cancer cells, increasing the heterogeneity among the cancer cells as they adapt to their unique local microenvironments [6,7,8,9]. Increased intratumoral heterogeneity has been shown to be associated with poorer patient prognosis and presents a difficult challenge in the treatment of cancer. Specifically in breast cancer, heterogeneity of cancer cell populations, accompanied with non-uniform drug delivery, can yield a diverse cellular response to treatment and result in clonal selection and an altered genomic landscape over the course of therapy [10,11,12,13]. Additionally, heterogeneous vasculature can lead to the emergence of hypoxic niches in regions deprived of oxygen, which have been associated with breast cancer resistance to chemo- and radiation therapies [14,15]. Characterizing intratumoral heterogeneity and its spatiotemporal evolution in response to both targeted and non-targeted therapies could yield better understanding of a patient’s individual tumor physiology and provide guidance for personalized treatment. 

Noninvasive imaging enables longitudinal quantification of physiological tissue properties through methods such as diffusion-weighted (DW) magnetic resonance imaging (MRI) and dynamic contrast-enhanced (DCE) MRI. DW-MRI depends on the mobility of water within biological tissue, allowing for noninvasive evaluation of tissue microstructure [16]. The apparent diffusion coefficient (*ADC*), calculated from DW-MRI data, has been correlated to tumor cellularity and cell membrane permeability [17], and increases in *ADC* values have been shown to be early markers of breast tumor therapeutic response [16,18]. DCE-MRI provides measures of vascular perfusion and permeability, as well as tissue volume fractions [19]. Pharmacokinetic parameters, *K^trans^* (i.e., the volume transfer constant) and *k_ep_* (efflux constant), derived from DCE-MRI data, can be used to quantify vascular alterations in response to treatment and serve as early markers of treatment response [20,21]. These quantitative imaging techniques, quantified on a voxel-wise basis, can characterize and provide biological insights into intratumoral heterogeneity across the entire tumor volume. 

Approaches to quantify intratumoral heterogeneity from imaging data include histogram and texture analysis [22]. Histogram analysis involves quantitative assessment of parameter map distributions, demonstrating value over whole-tumor summary statistics [23,24], and allowing for the quantification of longitudinal alterations in intratumoral heterogeneity [25,26]. However, this approach forgoes the spatial information provided in imaging data. To overcome the limitations of one-dimensional analyses, texture analysis provides measures of spatial heterogeneity that have demonstrated utility in the prediction of breast cancer treatment response [27], as well as the classification of benign and malignant lesions [28]. However, this method quantifies spatial variations across the whole tumor, assuming the tumor is heterogeneous, but “well-mixed” [6]. This differs from clinical image data, which has shown tumor heterogeneity presenting regionally as spatially distinct subregions [29,30,31]. In an effort to more comprehensively characterize intratumoral biological heterogeneity, recent work has investigated the segmentation of tumors into distinct subregions by clustering similar voxels from multiparametric image data [32,33,34,35]. This emerging approach has been designated as habitat imaging [36], as the imaging-derived subregions are representative of tumor “habitats”, akin to ecological habitats in which a particular species population resides. 

In this contribution, we investigated the ability of multi-parametric, voxel-based characterizations of tumor heterogeneity from DW- and DCE-MRI data to spatially resolve physiologically distinct tumor habitats in preclinical models of breast cancer. Two xenograft models of breast cancer were used, one a model of human epidermal growth factor receptor 2 positive (HER2+) disease (BT-474), and the other of triple negative disease (MDA-MB-231). We then utilized the identified habitats to quantify the longitudinal alterations of the tumor microenvironment in response to clinically relevant therapeutics. Finally, using end-point histology data, we biologically validated the imaging-derived habitats, through correlation with the habitats identified in the histological images.

## 2. Results

### 2.1. Discovery of MRI Tumor Habitats

Figure 1 presents the study design for the discovery of tumor habitats from quantitative MRI data and the analysis of longitudinal tumor composition defined by these habitats. The clustering analysis of quantitative imaging data yielded three primary clusters, or habitats, for both the BT-474 and MDA-MB-231 tumor cohort. The resultant dendrogram from agglomerative clustering (Appendix A), along with gap statistic plots were used to select a cluster number (*k*) of 3 (among two to 10 clusters) for both tumor cohorts. 

The mean parameter values (*K^trans^*, *k_ep_*, *ADC*, and *v_e_*) for each cluster were then evaluated to determine the physiological habitat each cluster represented. Statistically significant differences were found between the three identified clusters for all parameter values, in both the BT-474 (Figure 2a) and MDA-MB-231 (Figure 2b) tumor cohort, indicating that the agglomerative clustering algorithm was robust in identifying physiologically-distinct tumor habitats across the two preclinical models of breast cancer. 

To identify the physiology each cluster represented, the quantitative measures of vascularity, *K^trans^* and *k_ep_*, and cellularity, *ADC* and *v_e_,* were considered. For the BT-474 cohort (Figure 2a), the first identified cluster had a mean *K^trans^* of 0.18 min^−1^ and a mean *k_ep_* of 0.84 min^−1^, and was thus labeled as a “high-vascularity” (HV) habitat. In comparison, the second and third clusters had mean *K^trans^* values of 0.05 and 0.04 min^−1^, respectively, and mean *k_ep_* values of 0.31 and 0.08 min^−1^, respectively, and were labeled as “low-vascularity” (LV) habitats. Similar differences between clusters were observed in the MDA-MB-231 cohort (Figure 2b). The first cluster, labeled a “high-vascularity” (HV) habitat, had a mean *K^trans^* of 0.13 and a mean *k_ep_* of 0.77, while the second and third clusters had mean *K^trans^* values of 0.04 and 0.01, respectively, and mean *k_ep_* values of 0.32 and 0.01, respectively, and correspondingly were labeled as “low-vascularity” (LV) habitats. 

Next, quantitative measures of cellularity, *ADC* and *v_e_*, were considered. For the BT-474 cohort (Figure 2a), the third identified cluster had a mean *ADC* of 1.2 × 10^−3^ mm^2^/s and a mean *v_e_* of 0.78 and was thus labeled as a “low-cellularity” (LC) habitat. In comparison, the first and second clusters had mean *ADC* values of 6.9 × 10^−4^ and 6.2 × 10^−4^ mm^2^/s, respectively, and mean *v_e_* values of 0.27 and 0.18, respectively, and were labeled as “high-cellularity” (HC) habitats. Similar differences between cluster values were observed in the MDA-MB-231 cohort (Figure 2b) for the *v_e_* parameter. The third cluster had a mean *v_e_* of 0.99, while the first and second clusters had mean *v_e_* values of 0.27 and 0.17, respectively. While each cluster showed statistically significant differences in mean *ADC* for the MDA-MB-231 cohort, all three clusters had *ADC* values within the “high-cellularity” range (i.e., less than 1 × 10^−3^ mm^2^/s, See Materials & Methods) with values of 6.9 × 10^−4^, 5.1 × 10^−4^, and 5.7 × 10^−4^ mm^2^/s for the first, second, and third cluster, respectively. Taken with *v_e_* measures of cellularity, the third cluster was labeled as a “low-cellularity” (LC) habitat, while the first and second clusters were labeled as “high-cellularity” (HC) habitats. Altogether, the three identified clusters from both tumor cohorts represented the following physiological habitats: high-vascularity high-cellularity (HV-HC), low-vascularity high-cellularity (LV-HC), and low-vascularity low-cellularity (LV-LC). 

Figure 2c shows a representative habitat map and the corresponding quantitative parameter maps for a BT-474 tumor at day 0. Although no spatial information was used in the clustering process, the identified clusters were observed to colocalize spatially, demonstrating that we were able to spatially resolve physiologically distinct tumor habitats. These qualitative observations of spatial colocalization were confirmed quantitatively through multiregional spatial interaction (MSI) matrix analysis (Appendix A). Habitat voxels mapped to their original spatial index showed statistically higher MSI values (*p* < 0.01) than if mapped to a random index, confirming that the identified habitats colocalize spatially. 

### 2.2. Longitudinal Alterations in Tumor Composition in Response to Therapy 

Figure 3a shows representative tumor habitat maps for a BT-474 control and trastuzumab-treated tumor, on days 0, 1, and 4. Prior to treatment, at day 0, no significant differences (*p* > 0.05) were observed in tumor composition between treated and control tumors, with mean percent tumor volumes (and 95 % confidence interval) of 43.5 ± 10.9% and 29.3 ± 17.1%, for the HV-HC habitat, 32.1 ± 7.7% and 41.8 ± 11.1%, for the LV-HC habitat, and 24.4 ± 6.3% and 28.9 ± 13.6% for the LV-LC habitat, respectively (Figure 3b). At day 4, trastuzumab-treated tumors showed a significant increase in percent tumor volume of the HV-HC habitat compared to baseline, with a mean value of 66.9 ± 24.3% (*p* = 0.02), and a significant decrease in percent tumor volume of the LV-HC habitat compared to baseline, with a mean value of 9.6 ± 8.6% (*p* < 0.01). For control tumors, no significant changes from baseline in percent tumor volume of any habitat were observed over the course of the study (*p* > 0.05). At day 4, trastuzumab-treated tumors were observed to have significantly higher HV-HC percent tumor volume compared to control tumors (33.3 ± 19.3%, *p* = 0.03), and significantly lower LV-HC percent tumor volume compared to control (47.8 ± 29.4%, *p* = 0.03). No significant differences were observed for the LV-LC habitat between treated and control tumors (*p* > 0.05). 

Figure 3c shows representative tumor habitat maps for an MDA-MB-231 control and low-dose albumin-bound paclitaxel (ABP) treated tumor, at day 0, 1, and 3. Prior to treatment, at day 0, no significant differences (*p* > 0.05) were observed in tumor composition between low-dose ABP, high-dose ABP, and control tumors, with mean percent tumor volumes of 19.0 ± 6.4%, 25.0 ± 10.5%, and 26.1 ± 9.7% for the HV-HC habitat, 60.5 ± 5.7%, 54.3 ± 7.1%, and 57.1 ± 9.6% for the LV-HC habitat, and 20.5 ± 4.4%, 20.6 ± 4.6%, and 16.8 ± 5.7% for the LV-LC habitat, respectively (Figure 3d). Low-dose ABP-treated tumors showed a significant decrease from baseline in mean percent tumor volume of the LV-HC habitat at day 1 (45.2 ± 8.32%, *p* < 0.01) and at day 3 (49.5 ± 7.1%, *p* = 0.01). A corresponding nonsignificant increase in the percent tumor volume of the HV-HC habitat was observed at day 1 (32.8 ± 14.6%, *p* = 0.12) and at day 3 (27.4 ± 11.0%, *p* = 0.15). Control and high-dose ABP-treated tumors did not show any significant changes from baseline in percent tumor volume of any habitat over the course of the study (*p* > 0.05). Additionally, no significant differences were observed between treatment groups for any habitat over the course of the study (*p* > 0.05).

### 2.3. Discovery of Histological Tumor Habitats

Figure 1 shows the overall study design for the discovery of tumor habitats from histological data, through the use of stain density maps, and the biological validation of MRI-derived habitats using histological habitats. The clustering analysis of histological stain density maps yielded three primary clusters, or habitats, for both the BT-474 and MDA-MB-231 tumor cohort. For each cohort, a *k* = 3 showed the optimal gap statistic among 2 to 10 clusters. 

The mean vessel count and nuclei count for each cluster was then evaluated to determine the physiological habitat each cluster represented. Statistically significant differences were found between the three identified clusters for all stain density parameter values, in the BT-474 (Figure 4a) and MDA-MB-231 (Figure 4b) tumor cohort. For both tumor cohorts, the first identified cluster was labeled as “high-vascularity” (HV), with mean vessel counts/unit (in this context, “unit” refers to the 150 μm × 150 μm neighborhood around a pixel) of 5.5 for BT-474 cohort and 3.9 for MDA-MB-231 cohort. The second and third clusters showed mean vessel values of 2.2 and 0.7 counts/unit for the BT-474 cohort, and 0.58 and 0.28 counts/unit for the MDA-MB-231 cohort, respectively, and were labeled as “low-vascularity” (LV) habitats. For both tumor cohorts, the third identified cluster was labeled as “low-cellularity” (LC), with mean nuclei counts/unit of 17.8 for the BT-474 cohort and 60.1 for the MDA-MB-231 cohort. The first and second clusters showed mean nuclei values of 142.3 and 109.5 counts/unit, for the BT-474 cohort, and 249.9 and 220.8 counts/unit for the MDA-MB-231 cohort, respectively, and were labeled as “high-cellularity” (HC) habitats. Altogether, the three identified clusters from both tumor cohorts represented the following histological habitats: high-vascularity high-cellularity (HV-HC), low-vascularity high-cellularity (LV-HC), and low-vascularity low-cellularity (LV-LC).

Figure 4c shows a representative habitat map and the corresponding stain density maps for an example MDA-MB-231 tumor, from the low-dose ABP treatment group. As with the discovery of MRI habitats, while no spatial information was used in the clustering process to identify histological habitats, the identified clusters were observed to colocalize spatially, which was confirmed quantitatively through MSI matrix analysis (Appendix A). 

To further investigate the underlying physiologies that the identified histological habitats represent, we evaluated the necrosis and Ki-67 expression levels for each habitat. The LV-LC histological habitat showed high levels of necrosis (Figure 4), with a mean necrosis value of 48.1 percent area/unit for the BT-474 cohort and 21.1 percent area/unit for the MDA-MB-231 cohort. In comparison, the high cellularity habitats, HV-HC and LV-HC, showed low levels of necrosis with mean necrosis values of 9.4 and 10.9 percent area/unit for the BT-474 cohort, and 4.3 and 3.6 percent area/unit for the MDA-MB-231 cohort, respectively. The HV-HC histological habitat showed high levels of Ki-67 expression (Figure 4), with mean Ki-67+ nuclei values of 31.0 counts/unit for the BT-474 cohort and 23.7 counts/unit for the MDA-MB-231 cohort. In comparison, the low vascularity habitats, LV-HC and LV-LC, showed low levels of Ki-67 expression with mean Ki-67+ nuclei values of 6.6 and 0.5 counts/unit for the BT-474 cohort, and 8.5 and 0.9 counts/unit for the MDA-MB-231 cohort, respectively. 

### 2.4. Correlation between MRI-Derived and Histology-Derived Habitats

A positive linear correlation was observed between MRI percent tumor volume and histological percent tumor area of HV-HC habitats (Figure 5) for the BT-474 tumor cohort, with an *r^2^* = 0.74 (*p* = 0.03), and a nonsignificant positive correlation for the MDA-MB-231 cohort, with an *r^2^* = 0.23 (*p* > 0.05). A positive linear correlation was also observed between MRI percent tumor volume and histological percent tumor area of LV-LC habitats (Figure 5) for both the BT-474 tumor cohort, with an *r^2^* = 0.70 (*p* = 0.04), and the MDA-MB-231 cohort, with an *r^2^* = 0.73 (*p* < 0.01). Correlation analysis of the LV-HC habitats showed a nonsignificant positive correlation with an *r^2^* = 0.55 (*p* = 0.09), for the BT-474 cohort, and a significant positive correlation with an *r^2^* = 0.53 (*p* = 0.04), for the MDA-MB-231 cohort.

## 3. Discussion

In this contribution, we utilized quantitative imaging parameters derived from DW- and DCE-MRI to spatially resolve tumor habitats in two preclinical models of breast cancer. Three habitats were identified and labeled based on imaging measures of cellularity and vascularity as HV-HC, LV-HC, and LV-LC. Although no spatial information was used in the clustering process, identified habitats were found to colocalize spatially (Appendix A), demonstrating that the clustering of quantitative imaging parameters alone was robust to identifying tumor habitats. 

Interestingly, the HV-HC subregions were often observed to localize towards the outer edge of the tumor (Figure 2c), characteristic of the “enhancing rim” phenotype often empirically noted in tumors. Conversely, the LV-LC subregions were observed to localize towards the center of the tumor, possibly representing core necrosis frequently found in tumors. These habitat physiologies identified with MRI were biologically validated by comparing to clusters identified in corresponding histology data (Figure 5). Independent clustering of stain density maps yielded three histological habitats analogous to those identified from MRI data (HV-HC, LV-HC, and LV-LC), lending support to the selection of a *k* = 3 in the MRI habitat analysis. In both the BT-474 and MDA-MB-231 cohorts, the percent tumor area for all three histological habitats showed a positive linear correlation with the percent tumor volume of their corresponding MRI-derived habitats, (Figure 5b). This correlation indicates that tumor habitats derived from MRI data represent biologically distinct tumor subregions, defined in terms of cellularity and vascularity. Further analysis of histological habitats showed higher counts of Ki-67 positive nuclei in the HV-HC habitat compared to LV-HC and LV-LC habitats (Figure 4), suggesting that this habitat may represent viable tumor regions. These results align with findings by others of higher Ki-67 expression in regions of vascular perfusion at the tumor periphery [37,38]. Additionally, the LV-LC histological habitat showed significantly higher levels of necrosis compared to the HV-HC and LV-HC habitats (Figure 4), indicating this habitat may represent regions of necrosis, an observation supported by the literature [37]. It is noteworthy that habitat trends in mean parameter values (e.g., *K^trans^* of HV-HC > *K^trans^* of LV-HC > *K^trans^* of LV-LC) were consistent between the two tumor models, with the exception of *ADC* for the LV-LC habitat. The LV-LC habitat showed increased mean *ADC,* typical of necrosis [16,17], compared to other habitats for the BT-474 cohort (Figure 2a). However, the same was not observed for the MDA-MB-231 cohort (Figure 2b), a finding which was surprising since the imaged LV-LC habitat showed a significant correlation to the histological LV-LC (necrosis) habitat (Figure 5b). The LV-LC mean *ADC* for the MDA-MB-231 cohort (5.7 x 10^−4^ mm^2^/s) was similar to that observed for regions of necrosis in other studies of the same tumor model [39,40]. Differences in BT-474 and MDA-MB-231 necrosis morphology may attribute to differences observed in *ADC* values of the LV-LC habitat. 

The longitudinal evaluation of tumor habitat composition was facilitated by the use of quantitative imaging parameters, allowing the pooling of voxel data across imaging time points for each tumor cohort prior to clustering analysis. Clustering cohort voxel data, as opposed to individual tumor data, enabled the habitat analysis to be robust to inter-tumoral variations, avoiding the assumption that all habitats are present in each tumor, and ensure the habitat definitions were consistent across time. Prior to treatment, no difference in tumor composition was observed between treatment groups in either cohort (Figure 3). However, tumor composition at baseline differed between the HER2+ and triple negative tumors, with the BT-474 tumor cohort showing a higher percent tumor volume of the HV-HC habitat and lower percent tumor volume of the LV-HC habitat compared to MDA-MB231 (Figure 3b,d and Appendix A). The differences between subtypes in baseline tumor composition may be due to HER2’s involvement in VEGF expression and vascular recruitment [41], leading to potentially different vascular patterns between these two tumor models. Several radiomics studies have demonstrated the utility of clinical image features to predict molecular subtypes of breast cancer [42,43,44,45]. The observed differences in tumor composition at baseline indicate the potential to discriminate breast cancer subtypes using identified habitats.

In this study, habitats of therapeutic response were identified for both targeted and chemotherapies. At day 4 post-treatment with the anti-HER2 targeted therapy, trastuzumab, BT-474 tumors showed a significant decrease in the percent tumor volume of the LV-HC habitat (*p* < 0.01) and corresponding increase in the percent tumor volume of the HV-HC habitat (*p* = 0.02) from baseline. The LV-HC habitat represents a subregion with decreased vascular perfusion and higher cellularity, potentially corresponding to hypoxic regions with continued cell proliferation. The decrease in percent tumor volume of the LV-HC habitat upon trastuzumab treatment is concordant with findings from previous work, where trastuzumab therapy led to a longitudinal decrease in hypoxia in BT-474 tumors, quantified by ^18^F-FMISO positron emission tomography (PET) and pimonidazole staining of histological sections [46]. Observed increases in percent tumor volume of the HV-HC habitat and decreases in the LV-HC habitat upon trastuzumab treatment are supported by other studies demonstrating improved vascular perfusion resulting from trastuzumab therapy [47,48,49,50,51]. MDA-MB-231 tumors were treated with a low or high dose of albumin-bound paclitaxel (ABP), an established taxane cell-cycle inhibitor [52]. At day 3, the low dose of ABP therapy showed significant decreases in percent tumor volume of the LV-HC habitat (*p* = 0.01) compared to baseline values and corresponding increases (*p* > 0.05) in percent tumor volume of the HV-HC habitat. Paclitaxel has been shown to have anti-angiogenic effects [53,54], and its albumin-bound form has been shown to induce vascular remodeling [55]. Decreases in the LV-HC habitat upon low-dose ABP therapy may indicate improved vascular perfusion from the vascular remodeling effects of ABP therapy. 

Regions of low vascular perfusion and high cellularity, as with the LV-HC habitat, may potentially represent habitats where cells have adapted to low oxygen and nutrient conditions, resisting apoptosis and continuing to proliferate [6]. Thus, identifying these cell-dense regions of hypoxia may be of clinical interest as they could represent regions where underlying cell populations are more likely to be resistant to therapeutic interventions [5,15]. The LV-HC habitat may represent regions deprived of oxygen, although hypoxia was not directly measured in this study. However, several groups have evaluated the relationship between MRI-derived habitats and hypoxia, and found regions of moderate perfusion to often correspond to hypoxia [37,39,56]. Using spatially coregistered histology and multiparameteric MRI data, Jardim-Perassi et al. found MRI habitats corresponding to viable-hypoxic tissue showed decreased area under the curve values calculated from DCE-MRI, compared to viable-normoxic habitats [39]. Others have clustered DCE-MRI signal intensity time courses to identify subregions of different contrast agent uptake patterns and related identified subregions to tissue oxygenation. Stoyanova et al. found regions with delayed contrast enhancement and wash out to correspond to hypoxia, as measured by pimonidazole staining and ^18^F-FMISO-PET imaging [57]. Regions of similar enhancement patterns were identified by Featherstone et al., who found such regions to correspond with decreased oxygen enhancement as measured by oxygen-enhanced MRI [33]. 

Often quantitative imaging-based evaluations of treatment response involve summarizing an entire tumor volume down to a single statistic, largely ignoring the spatial heterogeneity within a tumor. In this study, we demonstrated the utility of habitat imaging to evaluate longitudinal alterations in tumor heterogeneity through the identification of tumor habitats of response to both targeted and non-targeted treatments. Other groups have also investigated the application of quantitative MRI parameter maps to identify tumor subregions of response. Quantitative *ADC* and *T_2_* maps have been used to segment tumor subregions of viable tissue and necrosis, with research showing significant changes in viable tumor fractions in response to anti-angiogenic [58] and radiation [59] therapy in preclinical tumor models. Longo et al. found analysis of tumor subregions derived from quantitative DCE-MRI parameters outperformed whole tumor analysis, when evaluating response to antiangiogenic therapy in a murine model of breast cancer [60]. These studies, accompanying our own, demonstrate the potential application of quantitative habitat imaging to measure the spatiotemporal evolution of intratumoral heterogeneity in response to therapy.

One limitation to our study was the lack of spatial registration of ex vivo histological habitats to those identified from in vivo imaging data. While such registration could provide a more direct biological validation to our in vivo findings, tissue section alignment with imaged slices, along with tissue shrinkage or deformation post-tumor excision, presents a difficult challenge for spatial registration of histology data to MRI [61]. While we were unable to spatially correlate imaged habitats with those derived from histology, we found significant correlations between tumor composition of histology and central-slice MRI data to lend biological support for MRI-derived habitats. Another important limitation is that identified habitats are constrained to the imaging parameters used to identify them, as well as the spatial resolution of the MRI acquisition. If additional imaging measures were employed, such as those sensitive to pH, hypoxia, or mechanical stiffness, it is possible that additional physiologically distinct habitats may be discovered. The spatial resolution of MRI also influences the extent of partial volume effects, as multiple habitats may be averaged within a tumor voxel [39]. Employing soft clustering techniques that provide a probabilistic cluster assignment to each voxel, such as gaussian mixture modeling, may be more robust to multiple habitats within a voxel and address these limitations. Finally, the imaging parameter thresholds used to label the MRI habitats as high or low cellularity or vascularity were selected manually, and future efforts will involve more systematic methods to determine these thresholds.

To apply these methods in the clinical setting, several challenges must be identified and then overcome. These include the limited signal-to-noise ratio and spatial and temporal resolution, as well as the presence of image artifacts (due to, for example, patient motion or biopsy clips), and the ability to obtain high-quality quantitative DW- and DCE-MRI parametric maps. The limited spatial resolution available in clinical MRI, may particularly confound this habitat imaging approach, as it is predicated upon being able to resolve and quantify intratumoral heterogeneity. Fortunately, clinical studies have shown that DW- and DCE-MRI metrics can be acquired with repeatability and reproducibility in the clinical setting [62,63,64], providing evidence of feasibility for the future application of our technology to clinical studies. In particular, the individual quantitative parameters derived from DW- and DCE-MRI have been identified as potential biomarkers of (for example) therapeutic response for breast cancer patients undergoing neoadjuvant therapy [16,19,20,65], thereby providing support for potential clinical translation of our habitat imaging approach. Furthermore, habitat imaging has already made significant contributions in other disease locations, such as in the brain [66,67] and prostate [68,69]; thus, it is a reasonable hypothesis that these methods may be applied for the predicting response in breast cancer patients.

## 4. Materials and Methods 

### 4.1. Cell Culture & Animal Model

Two xenograft tumor models were employed using the human-derived BT-474 (HER2+ breast cancer) and MDA-MB-231 (triple negative breast cancer) breast cancer cell lines (ATCC, Manassas, VA, USA). BT-474 cells were cultured in improved minimal essential medium (Invitrogen, Carlsbad, CA, USA) supplemented with 10% fetal bovine serum and 1% insulin at 37 °C in 5% CO_2_. MDA-MB-231 cells were cultured in Dulbecco’s modified essential medium supplemented with 10% fetal bovine serum at 37 °C in 5% CO_2_.

Data from two independent cohorts of mice as part of previous studies were used in this study [50,70]. Cohort one of female athymic nude mice were subcutaneously implanted with a 0.72 mg, 60-day release, 17β-estradiol pellet (Innovative Research of America, Sarasota, FL, USA) 24 h prior to subcutaneous injection of BT-474 cells. The second cohort of female athymic nude mice were subcutaneously injected with MDA-MB-231 cells. Tumors were grown until they reached approximately 275 mm^3^ in volume, when they were then randomly assigned to treatment groups. The BT-474 xenograft tumor cohort (*N* = 20) were either treated with trastuzumab (10 mg/kg) or saline control. The MDA-MB-231 xenograft tumor cohort (*N* = 37) were either treated with 15 mg/kg (low-dose) albumin-bound paclitaxel (ABP), 25 mg/kg (high-dose) ABP, or saline control. Treatment and imaging schedules for both cohorts of mice are shown in Figure 6. A flow chart of sample number for each treatment group, over the course of the study, is shown in Appendix A.

### 4.2. Magnetic Resonance Imaging and Analysis

While details are provided elsewhere [50,51,70], here we briefly summarize the salient MRI procedures. Mice were imaged using a 7T small animal MRI scanner (Agilent Technologies, Palo Alto, CA, USA) to acquire high resolution *T_2_*-weighted anatomical images, DW- and DCE-MRI. The day prior to the baseline imaging, a jugular catheter was implanted into mice for exogenous delivery of the contrast agent for DCE-MRI. All MRI data were acquired with transaxial slices, 1 mm slice thickness, and a 28 × 28 × 15 mm field of view. High resolution *T_2_*-weighted anatomical images were collected for tumor segmentation using a 128 × 128 × 15 matrix and *TR*/*TE*/NEX = 5500 ms/ 35.65 ms/2. Tumor regions-of-interest (ROIs) were manually drawn around the tumor boundary for all slices containing the tumor using the high resolution *T_2_*-weighted images and custom-built MATLAB scripts (MATLAB, MathWorks, Natick, MA, USA). ROIs were modified manually to exclude regions where no *T_1_*-weighted or *T_2_*-weighted signal was observed, suggestive of calcifications [71]; see Appendix A. 

Diffusion-weighted data was collected using a standard pulse gradient spin echo sequence with three *b*-values (150, 500, and 800 s/mm^2^) and the following scan parameters: 64 × 64 matrix, *TR*/*TE*/NEX = 2000 ms/30ms/2; gradient duration and intervals of *δ* = 3 ms and *Δ* = 20 ms, respectively. The signal intensity data from acquired DW-MRI was fit voxel-wise (MATLAB) to Equation (1) to calculate ADC maps via: (1)$$S\left(b\right)=S_{0}\exp\left(-ADC\times b\right),$$ where *S*(*b*) is the signal intensity after gradient application, *S_0_* is the signal intensity prior to gradient application, and *b* is the strength of the diffusion gradient. Voxels within the ROI were removed from analysis if the signal intensities at the three *b*-values did not decrease monotonically; i.e., we require that *S*(*b* = 800) < *S*(*b* = 500) < *S*(*b* = 150). The average (and 95% confidence interval) percent of tumor voxels removed was 7.31% (1.30%) per DWI scan. Slices of the DW-MRI data showing motion artifacts were removed from subsequent analysis if greater than 20% of the slice’s voxel were removed via the aforementioned criteria. DW-MRI data for a tumor on a given imaging time point was discarded entirely if more than half of the slices were removed from analysis due to the aforementioned criteria (BT-474 cohort: 6 out of 56 scans, MDA-MB-231 cohort: 3 out of 110 scans, Appendix A).

After DW-MRI acquisition, pre-contrast *T_1_* maps were acquired using a multislice inversion recovery snapshot fast low angle shot gradient echo sequence. Seven inversion times ranging from 250 to 10,000 ms were used in addition to the following scan parameters: 64 × 64 matrix, *TR*/*TE*/NEX = 12,000 ms/2.1 ms/2, and a flip angle of *α* = 15 °. *T_1_* maps were calculated by fitting voxel data (MATLAB) to Equation (2): (2)$$S\left(TI\right)=S_{0}\left|\left(1-2\exp\left(\frac{-TI}{T_{1}}\right)+\exp\left(\frac{-TR}{TI}\right)\right)\right|,$$ where *S*(*TI*) is the signal intensity at inversion time *TI* and *S_0_* is the baseline signal intensity.

DCE-MRI acquisition involved the collection of dynamic *T_1_*-weighted images using a spoiled gradient echo sequence with a temporal resolution of 12.8 s for 20 minutes, with the following scan parameters: 64 × 64 matrix, *TR*/*TE*/NEX = 100 ms/2.1 ms/2, and a flip angle of *α* = 25 °. Pre-contrast images were acquired for approximately two minutes before a bolus of 0.05 mmol/kg gadolinium-diethylenetriaminepentaacetic acid (Magnevist, Bayer, Whippany, NJ) was delivered via a jugular catheter. DCE-MRI data were voxel-wise fit to the standard Kety-Tofts model (Equation (3) [72,73]:(3)$$C_{t}\left(t\right)=K^{trans}\int_{0}^{t}C_{p}\left(u\right)\exp\left(-\left(\frac{K^{trans}}{v_{e}}\right)\left(t-u\right)\right)du,$$ using a population-derived [74] arterial input function (AIF) [74,75], *C_p_*(*t*), to estimate *K^trans^*, and *v_e_* (i.e., the extravascular extracellular volume fraction). To reduce systematic errors in the model fits, the AIF was calibrated for each DCE-MRI scan to segmented skeletal muscle tissue [76] prior to fitting for pharmacokinetic parameters. Additionally, each tumor voxel time-course was fit to a half-logistic function to determine the bolus arrival-time, which was used to shift the calibrated AIF on a voxel-wise basis to align the enhancement time of *C_p_*(*t*) with the bolus arrival-time [77]. Voxel-wise fits to the standard Kety-Tofts model were bound within the physiological range for each parameter (e.g., 0 to 5 min^−1^ for *K^trans^*, 0 to 1 for *v_e_*). The efflux constant, *k_ep_*, was calculated as *K^trans^*/*v_e_*, with values beyond the physiological range (i.e., if 0 min^−1^ < *k_ep_* < 5 min^−1^) removed from analysis. The average (and 95% confidence interval) percent of tumor voxels removed was 0.76% (0.15%) per DCE scan. DCE-MRI data was not used if contrast agent delivery was unsuccessful (BT-474 cohort: four out of 56 scans, MDA-MB-231 cohort: 12 out of 110 scans, Appendix A).

### 4.3. Discovery of MRI Tumor Habitats

Multiparametric voxel data was extracted for tumor ROIs yielding a four-dimensional vector whose entries consist of *K^trans^*, *v_e_*, *k_ep_,* and *ADC* for each voxel. For each tumor cohort, voxel data was pooled across all imaging time points and each parameter distribution was scaled to have a mean of zero and standard deviation of one, to allow all parameters to contribute equally in the clustering process [78]. Tumor habitats were then identified through automated clustering of voxel data using an agglomerative clustering [79] algorithm with the Ward linkage and Euclidean distance measure (RStudio, “Rclusterpp” package). Details of agglomerative clustering are described in the supplemental methods. In this study, both the dendrogram branch height and gap statistic [80] were used to determine the number of clusters (RStudio, “cluster” package). No spatial information was used in the clustering process. 

Cluster-labeled voxels were then mapped back to their original spatial index and a multiregional spatial interaction (MSI) matrix (adapted from Wu et al. [81]) was used to quantify spatial colocalization of cluster-labeled voxels. Details of MSI matrix analysis are described in supplemental methods. To identify what type of physiology each cluster (habitat) represented, the mean value for each MRI parameter was calculated to label each identified cluster in terms of high or low “vascularity” or “cellularity”. Clusters with mean *K^trans^* > 0.1 min^−1^ or *k_ep_* values > 0.5 min^−1^ were designated as high-vascularity (HV) habitats, and otherwise as low-vascularity (LV) habitats. Clusters with mean *ADC* < 1 × 10^−3^ mm^2^/s or mean *v_e_* < 0.3 were designated as high-cellularity (HC) habitats, and otherwise as low-cellularity (LC) habitats.

### 4.4. Quantifying Longitudinal Alterations in Tumor Composition

Longitudinal alterations in tumor heterogeneity were quantified by changes in the percent of tumor volume comprised by each habitat to determine if habitat tumor composition changes with respect to a therapeutic intervention. The percent tumor volume of each habitat was averaged within each treatment group and evaluated at every time point for each tumor model cohort. Tumors that were larger than 500 mm^3^ at day 0 were excluded from treatment response analysis (BT-474: *N* = 1, MDA-MB-231: *N* = 2, Appendix A). 

### 4.5. Immunohistochemistry Staining and Image Processing

On the final imaging time point, (i.e., day 3 for the MDA-MB-231 cohort, and day 4 for the BT-474 cohort) each animal was sacrificed and the tumor excised for histological analysis (BT-474 cohort: *N* = 16, MDA-MB-231 cohort: *N* = 19, Appendix A). Tumors were sliced at the largest cross-section corresponding (as best as possible) to the central in vivo imaging plane and fixed in 10% formalin for 48 h, then stored in 70% ethanol for processing. Upon processing, tissue samples were embedded in paraffin and sliced into 5 μm-thick sections. Sections were mounted onto glass slides and stained with hemotoxylin and eosin (H&E), anti-CD31, or anti-Ki-67 as previously described [51,70]. A Leica SCN400 (Leica Biosystems, Nussloch, Germany) slide scanner was used to digitally scan slides in high-resolution (20×) brightfield. Images were processed using Bio-Formats [82] (http://openmicroscopy.org) and custom-built MATLAB scripts.

Images of histology sections were then automatically segmented to yield stain-positive masks across the entire tumor slice (Appendix A). A *k-*means clustering algorithm was used for automated segmentation of the H&E stained sections to generate masks for nuclei and necrosis across the entire tumor slice. Otsu thresholding [83] was used for automated segmentation of proliferative cell nuclei from anti-Ki-67 stained sections and microvessels from anti-CD31 stained sections. Further details of the automated segmentation techniques are described in previous work [26]. Masks of whole tissue for each stained tissue slice were semi-automatically generated using gray (non-tissue) area masks and manually drawn ROIs (Appendix A). These whole tissue masks were used to spatially align different histological stains from separate slides to the corresponding H&E stain for each tumor sample, through a rigid registration algorithm (Appendix A).

To quantify regional densities in cellularity and vascularity, a neighborhood analysis was performed to generate stain density maps for each tissue section. A 150 μm × 150 μm [84] neighborhood around each pixel was evaluated to facilitate comparison to the larger MRI voxels (437.5 μm × 437.5 μm) without losing the ability to assess regional variations of the tumor microenvironment. To quantify CD31+ microvessel density, H&E nuclear density, and Ki-67+ nuclear density, individual connected components within a 150 μm × 150 μm neighborhood around each pixel were counted across each tissue slice. Each neighborhood count was then assigned as that pixel’s nuclei (H&E, Ki-67) or microvessel (CD31) density value, thus yielding a neighborhood heatmap of stain density (Appendix A). A similar neighborhood analysis was performed for H&E necrosis density, where the percent stain-positive area within a 150 μm × 150 μm neighborhood around each pixel was calculated and assigned as that pixel’s necrosis density value (Appendix A). Stain density maps are presented in counts per unit (counts/unit) or percent area per unit (percent area/unit), where a “unit” represents the 150 μm × 150 μm neighborhood around each pixel.

### 4.6. Discovery of Histological Tumor Habitats

To reduce redundancy in the clustering analysis of stain density maps, each stain density map was downsampled by a factor of 0.01, using nearest neighbor interpolation (Appendix A). Multiparametric pixel data from spatially-aligned stain density maps were extracted for the tissue-masked areas of each tumor, yielding a four-dimensional vector for each pixel (nuclei, necrosis, Ki-67+ nuclei, and vessel density). Pixel data was pooled within each tumor cohort (BT-474 or MDA-MB-231) and each parameter distribution was scaled to have a mean of zero and standard deviation of one, so that all parameters contribute equally in the clustering process [78]. Pixel data was then clustered (Appendix A) through agglomerative clustering, in the same manner as MRI-derived habitats (described above), to identify physiologically-distinct histological subregions within tumor cross-sections. No spatial information was used in the clustering process. Tumors were included in the analysis only if all three (H&E, CD31, and Ki-67) stained sections were available (BT-474 cohort: *N* = 12, MDA-MB-231 cohort: *N* = 13, Appendix A).

Cluster-labeled pixels were then mapped back to their original spatial index and spatial colocalization of cluster-labeled pixels was measured through MSI matrix analysis, in the same manner as MRI-derived habitats. To identify what type of physiology each cluster (habitat) represented, the mean value for vessel and nuclei count was calculated to label each identified cluster in terms of high or low “vascularity” or “cellularity”, respectively. Clusters with mean vessel count/unit values greater than three were designated as high-vascularity (HV) habitats, and otherwise as low-vascularity (LV) habitats. Clusters with mean nuclei count/unit values greater than 100 were designated as high-cellularity (HC) habitats, and otherwise as low-cellularity (LC) habitats.

### 4.7. Correlations between MRI and Histological Habitats

To biologically validate MRI-derived habitats, a correlation analysis was performed between endpoint MRI and corresponding histology data. Tumors with both endpoint MRI data and histology data taken from the central tumor cross-section, were included in the analysis (BT-474 cohort *N* = 6, MDA-MB-231 cohort: *N* = 8, Appendix A). To identify central slices of the tumor ROI, high-resolution *T_2_*-weighted MRI and *b* = 800 DW-MRI data were visualized alongside corresponding H&E histology sections. Each of the MRI slices were stepped through and the anatomical similarity of the shape of the tumor slice as well as the presence and location of regions of necrosis were assessed to determine the closest possible match to the histology slice (Appendix A). The percent tumor volume of each habitat was calculated for the central slice of the tumor ROI at the study end point (day 4 for BT-474 cohort, day 3 for MDA-MB-231 cohort). These values were then correlated with the percent tumor area of each habitat across the histological section. 

### 4.8. Statistical Analysis

To test if differences in mean parameter values between habitats (MRI-derived or histology-derived) were significant, a one-way analysis of variance (ANOVA) followed by Tukey’s honest significant difference test was used. Differences from baseline within treatment groups were compared using a nonparametric Wilcoxon rank sum test. Differences between treatment groups were compared using a nonparametric Wilcoxon rank sum test (BT-474) or one-way ANOVA followed by Tukey’s honest significant difference test (MDA-MB-231). Correlations were tested using Pearson’s product-moment correlation. In all statistical tests a *p*-value less than 0.05 was considered significant.

## 5. Conclusions

This study employed quantitative MRI data to spatially resolve physiologically-distinct tumor habitats of varying cellularity and vascular properties in two preclinical models of breast cancer. We identified tumor habitats associated with therapeutic response to targeted and chemotherapies. We also provided biological validation of imaged habitats through correlation analysis with histological habitats identified from stain density maps. Application of habitat imaging techniques utilizing physiological parameters derived from quantitative MRI could provide greater understanding of a patient’s individual tumor physiology and provide guidance for personalized cancer treatment.

## Figures and Tables

**Figure 1 cancers-12-01682-f001:**
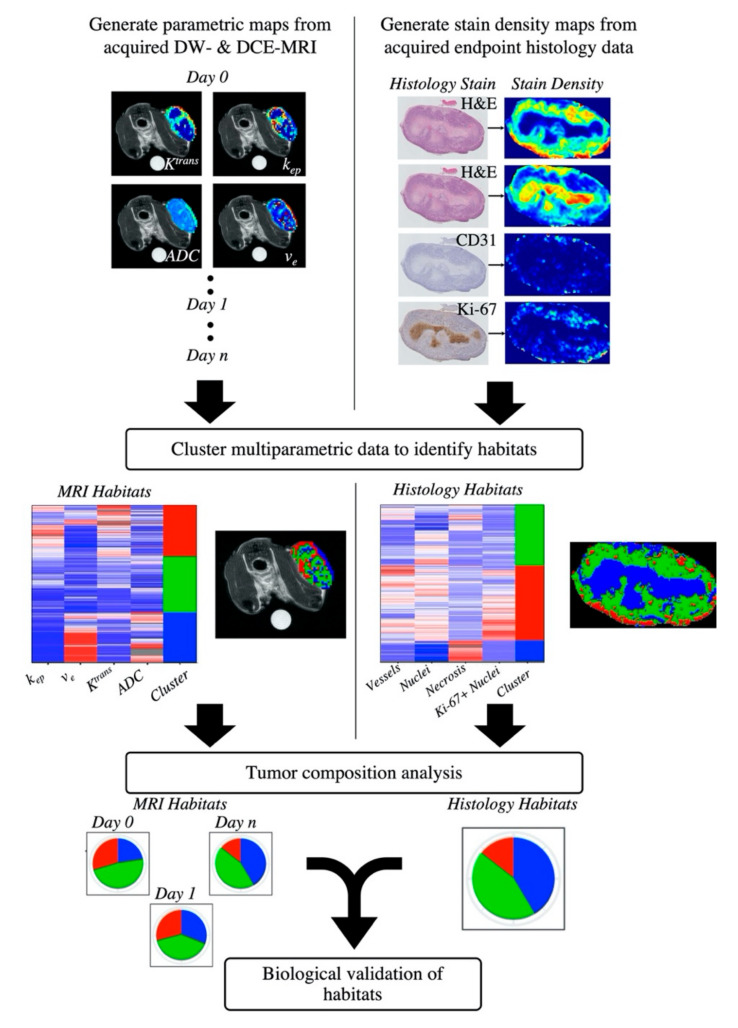
Pipeline for the habitat imaging analysis. Longitudinal DCE- and DW-MRI data was acquired for each animal and processed to generate individual multi-parametric maps of the tumor region (top row, left). Each tumor voxel was then defined as a vector where each element corresponded to one of the parameters returned from the quantitative MRI analysis. Voxel data within each individual tumor cohort (BT-474 or MDA-MB-231) were pooled together. The pooled voxel data is then clustered to identify tumor habitats (middle row, left). Tumor composition over time was quantified in terms of the percent of each tumor’s volume having a particular habitat (bottom row, left). End point histology data was acquired and processed to generate stain density maps (top row, right). Each pixel was then defined as a vector of its stain density values and pixel data within each tumor cohort were pooled together. The pooled pixel data is clustered to identify histological tumor habitats (middle row, right). Histological tumor composition was then quantified in terms of the percent of each tumor’s area having a particular habitat (bottom row, right). To biologically validate imaging-derived habitats, a correlation analysis was performed between habitat percent tumor volume from the imaging-derived habitats, and habitat percent tumor area from the histology-derived habitats.

**Figure 2 cancers-12-01682-f002:**
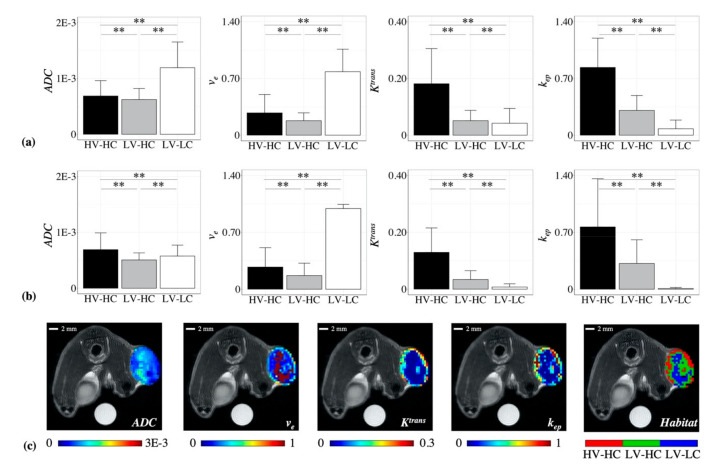
The mean parameter values for each of the identified habitats are shown for the (**a**) BT-474 and (**b**) MDA-MB-231 cohorts. The mean parameter values were used to identify the physiological habitat each cluster represented in terms of high or low “vascularity” or “cellularity”. The three identified habitats were as follows: high-vascularity high-cellularity (HV-HC), low-vascularity high-cellularity (LV-HC), low-vascularity low-cellularity (LV-LC). All three habitats were found in both the BT-474 and MDA-MB-231 cohorts. Error bars show standard deviation and ** indicates a *p* < 0.01. Panel (**c**) shows representative parameter maps and a corresponding habitat map for an example BT-474 tumor at baseline, where the HV-HC habitat is shown in red, LV-HC in green, and LV-LC in blue.

**Figure 3 cancers-12-01682-f003:**
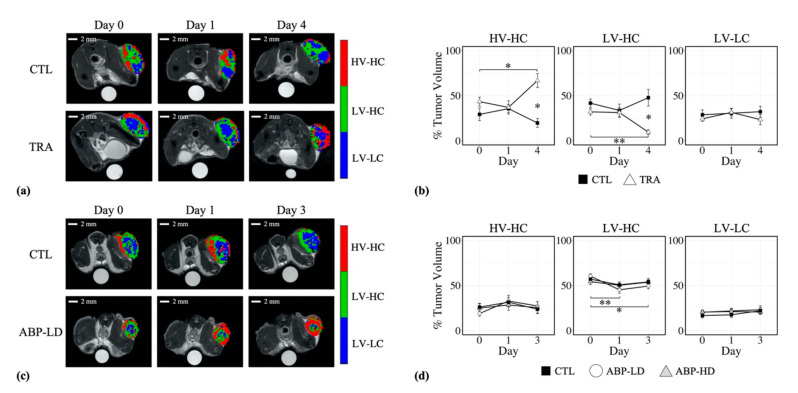
Panel (**a**) shows representative tumor habitat maps for a BT-474 control (CTL) and trastuzumab-treated (TRA) tumor on days 0 (pre-treatment), 1, and 4. Trastuzumab-treated tumors showed longitudinal increases in percent tumor volume (panel (**b**)) of the HV-HC habitat at day 4 (*p* = 0.02), and significantly higher percent tumor volumes of the HV-HC habitat compared to control (*p* = 0.03). Trastuzumab-treated tumors also showed longitudinal decreases in percent tumor volume (panel (**b**)) of the LV-HC habitat at day 4 (*p* < 0.01) and significantly lower percent tumor volumes of the LV-HC habitat compared to control (*p* = 0.03). Panel (**c**) shows representative tumor habitat maps for an MDA-MB-231 control (CTL) and low-dose albumin-bound paclitaxel (ABP-LD) tumor on days 0 (pre-treatment), 1, and 3. Low-dose ABP-treated tumors showed significant decreases in percent tumor volume (panel (**d**)) of the LV-HC habitat at day 1 (*p* < 0.01) and day 3 (*p* = 0.01). No significant changes were observed for control and high-dose ABP (ABP-HD) tumors over the course of the study. No significant differences were observed between treatment groups for the MDA-MB-231 cohort. Statistical significance is denoted with a * for *p* < 0.05 and ** for *p* < 0.01, error bars represent standard error.

**Figure 4 cancers-12-01682-f004:**
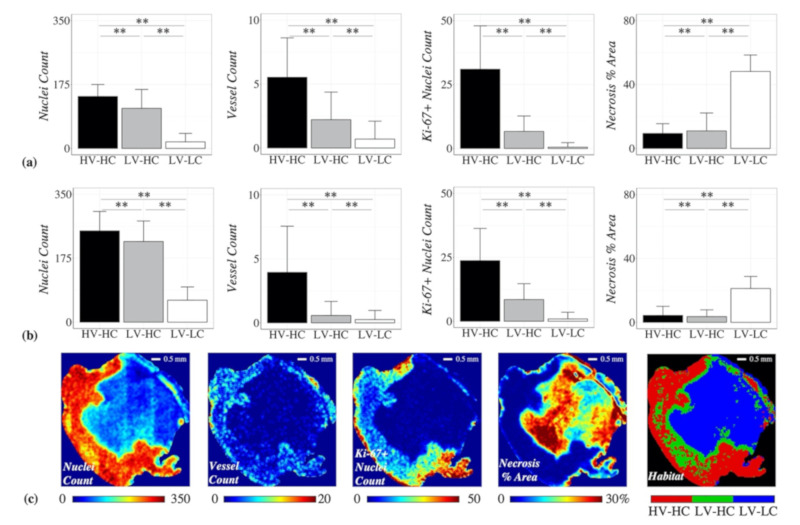
The mean stain density values for each of the histological habitats are shown for the (**a**) BT-474 and (**b**) MDA-MB-231 cohorts. The mean vessel count and nuclei count values were used to identify the physiological habitat each cluster represented in terms of high or low “vascularity” or “cellularity”, respectively. The three identified habitats were as follows: high-vascularity high-cellularity (HV-HC), low-vascularity high-cellularity (LV-HC), low-vascularity low-cellularity (LV-LC). All three habitats were found in both the BT-474 and MDA-MB-231 cohorts. Error bars show standard deviation and ** indicates a *p* < 0.01. Panel (**c**) shows representative stain density maps and a corresponding habitat map for an example MDA-MB-231 tumor from the low-dose ABP treatment group, where the HV-HC habitat is shown in red, LV-HC in green, and LV-LC in blue.

**Figure 5 cancers-12-01682-f005:**
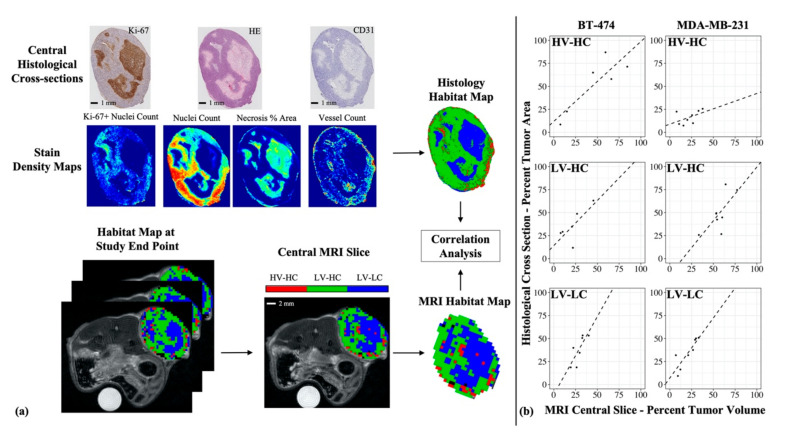
Correlation of MRI-derived tumor habitats with histological habitats. Stain density maps were used to generate histological habitat maps. The percent tumor area of each habitat was calculated for each histological section (panel (**a**)). Central imaging slices were selected from study endpoint (day 4 for BT-474 cohort, day 3 for MDA-MB-231 cohort) MRI data (panel (**a**)) and percent tumor volume of each habitat was calculated for each tumor’s central slice. The correlation was calculated between percent tumor volume (MRI) and percent tumor area (histology) for each habitat. For the BT-474 cohort (panel (**b**), left column), percent tumor volume of MRI-derived habitats showed a significant positive correlation with percent tumor area of histology-derived habitats for the HV-HC (*r^2^* = 0.74, *p* = 0.03) and LV-LC (*r^2^* = 0.70, *p* = 0.04) habitats. For the MDA-MB-231 cohort (panel (**b**), right column), percent tumor volume of MRI-derived habitats showed a significant positive correlation with percent tumor area of histology-derived habitats for the LV-HC (*r^2^* = 0.70, *p* = 0.04) and LV-LC habitats (*r^2^* = 0.73, *p* < 0.01).

**Figure 6 cancers-12-01682-f006:**
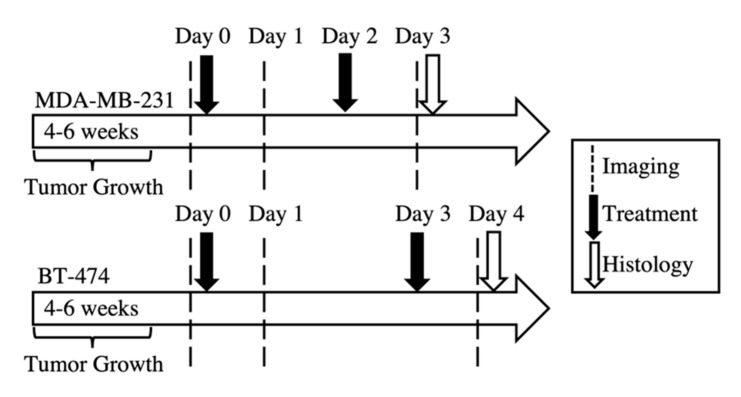
Tumor implantation, imaging, and treatment schedule for the tumor cohorts. MDA-MB-231 tumors were imaged on days 0, 1, and 3, while BT-474 tumors were imaged on days 0, 1, 4. MDA-MB-231 tumors were treated on day 0 (post-imaging) and day 2, with either saline (control), 15 mg/kg albumin-bound paclitaxel (ABP), or 25 mg/kg ABP treatment. BT-474 tumors were treated on days 0 (post-imaging) and 3, with either saline (control) of 10 mg/kg trastuzumab treatment.

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
