# Peer review of "Multiparametric Analysis of Longitudinal Quantitative MRI Data to Identify Distinct Tumor Habitats in Preclinical Models of Breast Cancer"

_cancers, 2020, doi:10.3390/cancers12061682_

Round 1
Reviewer 1 Report
In this contribution, the authors characterized tumor heterogeneity by using in vivo DW- and DCE-MRI data to spatially resolve physiologically distinct tumor habitats in two breast cancer xenograft models. Clustering of MRI data yielded three clusters high-vascularity high-cellularity (HV-HC), low-vascularity high-cellularity (LV-HC), and low vascularity low-cellularity (LV-LC). This clustering was validated by histology data through correlating percent tumor volume (MRI) to area (histology) of each habitat but not by spatial registration. A big selling point of this work is the use of the multiparametric MRI data to quantify the longitudinal alterations of the tumor microenvironment in response to clinically relevant therapeutics. This manuscript is well organized and written. The limitations are explicitly discussed as well. Other than one minor concern, I give my favorable recommendation.
To register MRI data with histology, the authors used central slice from MRI but no internal marker was used. I suggest the authors to elaborate how to select the central slice in MRI and histology for registration processes.
Reviewer 2 Report
The manuscript entitled "Multiparametric analysis of longitudinal quantitative MRI data to identify distinct tumor habitats in preclinical models of breast cancer" presents a methodology to identify physiological tumor habitats for two breast cancer models, using longitudinal MRI data.The authors present a well executed scientific study in extensive detail; however, there are a few questions / concerns that need to be addressed that will make this work more valuable.
- The authors mention that the spatial resolution of the MRI plays a key role in identifying habitats. Have the authors tried using a 3T or a 1.5T MR and compared the results to their 7T findings presented here ? With the current state of the art being 7T, it is difficult to obtain better spatial resolution than that; however, a comparative study using different scanners and different strengths can provide meaningful insights into the effect of resolution on identification of habitats. This would be an excellent addition to the manuscript.
- Line 436: ROIs are drawn manually around the tumor boundary. How are these ROIs drawn ? At the MR console using the imaging interface provided on the scanner ?
Additionally, what role does human error play in the final results? Since the ROIs are drawn manually, they are largely dependent on the observer and the size of the ROI as well as the exclusions will change depending on who performs the analysis. Have the authors controlled for this effect ? An easy way to account for the issue would be to draw multiple / repeat ROIs on each tumor to calculate the margin of error in the analysis.
This information is critical and could affect the results heavily. - Clinical application: Since the end goal of this work is to translate these methods to human patients, how do the authors anticipate doing that successfully ? Human patients pose a new set of challenges - motion during MRI, higher number of artifacts on the imaging due to various different factors, not adequate resolution for certain patients, not all clinics may have a 7T accessible for breast MR. Although the study presented here is novel and has potential, translating these methods clinically pose quite a few challenges. The authors need to address these questions in the Discussion section.
- Structure / Organization: Typically, manuscripts follow the Intro, Methods, Results, Discussion format to allow the reader to understand the project without going back and forth. Please reorganize / restructure the manuscript accordingly.
- Lines 448-450: Please reword this confusing sentence
- Line 580-581: Please reword this confusing sentence
Round 2
Reviewer 2 Report
The authors have addressed all the concerns raised in the prior round of reviews. I recommend this manuscript for publication.